# Accounting for Sampling Weights in the Analysis of Spatial Distributions of Disease Using Health Survey Data, with an Application to Mapping Child Health in Malawi and Mozambique

**DOI:** 10.3390/ijerph19106319

**Published:** 2022-05-23

**Authors:** Sheyla Rodrigues Cassy, Samuel Manda, Filipe Marques, Maria do Rosário Oliveira Martins

**Affiliations:** 1Department of Mathematics and Informatics, Faculty of Sciences, Eduardo Mondlane University, Maputo 254, Mozambique; sheylaratan@hotmail.com; 2Centre for Mathematics and Applications, CMA, NOVA School of Science and Technology, NOVA University of Lisbon, 2829-516 Lisbon, Portugal; fjm@fct.unl.pt; 3Department of Statistics, University of Pretoria, Pretoria 0028, South Africa; 4Biostatistics Research Unit, South Africa Medical Research Council, Pretoria 0001, South Africa; 5Department of Mathematics, NOVA School of Science and Technology, NOVA University of Lisbon, 2829-516 Lisbon, Portugal; 6Global Health and Tropical Medicine, GHTM, Instituto de Higiene e Medicina Tropical, IHMT, Universidade Nova de Lisboa, 1349-0008 Lisbon, Portugal; mrfom@ihmt.unl.pt

**Keywords:** survey sampling weights, disease mapping, child malnutrition, fever and diarrhea, Bayesian spatial smoothing, sub-Saharan Africa

## Abstract

Most analyses of spatial patterns of disease risk using health survey data fail to adequately account for the complex survey designs. Particularly, the survey sampling weights are often ignored in the analyses. Thus, the estimated spatial distribution of disease risk could be biased and may lead to erroneous policy decisions. This paper aimed to present recent statistical advances in disease-mapping methods that incorporate survey sampling in the estimation of the spatial distribution of disease risk. The methods were then applied to the estimation of the geographical distribution of child malnutrition in Malawi, and child fever and diarrhoea in Mozambique. The estimation of the spatial distributions of the child disease risk was done by Bayesian methods. Accounting for sampling weights resulted in smaller standard errors for the estimated spatial disease risk, which increased the confidence in the conclusions from the findings. The estimated geographical distributions of the child disease risk were similar between the methods. However, the fits of the models to the data, as measured by the deviance information criteria (DIC), were different.

## 1. Introduction

In epidemiology and public health, the methods for mapping disease have long been used to estimate spatial patterns of disease risk. Statistical advances in the methods have included spatial smoothing of disease risk to produce interpretable maps, and extensions to include temporal components as well as individual and geographical-level data. Estimation of geographical patterns of diseases in low-resource settings is increasingly important in guiding decision-making on where to allocate resources [1,2]. 

In sub-Saharan Africa, many disease mapping analyses that use data from complex health surveys fail to account for the survey designs such as disproportionate sampling [2]. In standard survey analyses, disproportionate sampling is corrected in the analysis by using the survey sampling weights that adjust for the disproportionate contribution of each ultimate sampling unit to the whole sample data. Ignoring the sampling weight in disease mapping analyses could lead to biased estimates of the spatial distributions of the disease risk, which could adversely affect policy decisions based on them. Thus, appropriate statistical analysis methods that incorporate sampling weights in the estimation of spatial patterns of diseases are critical [3,4,5,6,7].

In this paper, rather than detailing the epidemiology of child diseases in sub-Saharan Africa, a research topic that has extensively been analysed in several disease-mapping analyses in Africa, we present recent statistical analysis methods for incorporating sampling weights in the estimation of the geographical distributions of disease risk using complex health survey data. The two datasets: 2015-16 Malawi Demographic and Health Survey (2015-16 MDHS) [8], and the 2015 Mozambique Immunization Malaria and HIV/AIDS Key Indicator Survey (IMASIDA 2015) [9], are used for illustrative purposes using the mapping of child malnutrition, fever, and diarrhoea.

## 2. Methods

### 2.1. General Notation

Suppose that a finite population U={1,2,…,N} is distributed into i=1,2,…,I areas, and a random probability sample survey s with size n is taken according to a given design. Let $N_{i}$ and $n_{i}$ be the population and sample of sizes for area i, respectively, such that $U={\cup^{\;}}_{i=1}^{I}U_{i},\;N=\sum_{i=1}^{I}N_{i},\;s={\cup^{\;}}_{i=1}^{I}s_{i}\;\mathrm{and}\;n=\sum_{i=1}^{I}n_{i}$. Let $Y_{ij}$ be the binary indicator for the presence of the disease, taking a value of 1 or 0 on whether or not the jth individual has the disease in area $i\;\left(j=1,\ldots ,N_{i};\;i=1,\ldots ,I\right)$. It is assumed that $N_{i}$ is known for each area i. Further, we assume that individual ij has a known probability $\pi_{ij}$ of being included in the sample.

Our interest is to estimate the true area-specific population prevalence $P_{i}$, which is defined as: (1)$$P_{i}=\frac{1}{N_{i}}\overset{N_{i}}{\underset{j=1}{\sum }}Y_{ij},$$
using the accrued sample from area i. The area-specific unweighted estimator of the true area prevalence $P_{i}$ is given by ${\hat{P}}_{i}^{UW}$ which is calculated as: (2)$${\hat{P}}_{i}^{UW}=\frac{1}{n_{i}}\overset{n_{i}}{\underset{j=1}{\sum }}y_{ij},$$
and its variance is obtained as: (3)$$\hat{var}\left({\hat{P}}_{i}^{UW}\right)=\frac{{\hat{P}}_{i}^{UW}\left(1-{\hat{P}}_{i}^{UW}\right)}{n_{i}}.$$

In the case of a simple random sampling design without replacement, the estimator (2) is unbiased. However, in complex sampling, it would be inadequate as it does not account for the sample survey design, for example, sampling weights [10]. 

### 2.2. The Horvitz–Thompson Estimator

The well-known sample-design based unbiased estimator of the population prevalence $P_{i}$ is the Horvitz–Thompson (HT) estimator [11], which is given by: (4)$${\hat{P}}_{i}^{HT}=\frac{\sum_{j\in s_{i}}w_{ij}^{d}y_{ij}}{\sum_{j\in s_{i}}w_{ij}^{d}}=\frac{1}{n_{i}}\underset{j\in s_{i}}{\sum }{\tilde{w}}_{ij}^{d}y_{ij},$$
where $s_{i}$ is the set of individuals who are sampled from area i, with $y_{ij}$ being the observed value for $j\in s_{i}$ with $\left|s_{i}\right|=n_{i}$, $w_{ij}^{d}=\frac{1}{\pi_{ij}}$ the design weight (i.e., the sampling weights are the inverse probability of inclusion in the sample adjusted for non-response [12]), and ${\tilde{w}}_{ij}^{d}$ given by: (5)$${\tilde{w}}_{ij}^{d}=n_{i}\frac{w_{ij}^{d}}{\sum_{j\in s_{i}}w_{ij}^{d}},$$
is the normalized sampling weight. According to study [4], an estimator of the variance of ${\hat{P}}_{i}^{HT}$ can be expressed as follows:(6)$$\hat{var}\left({\hat{P}}_{i}^{HT}\right)=\frac{1}{n_{i}}\left(1-\frac{n_{i}}{N_{i}}\right)\frac{1}{n_{i}-1}\underset{j\in s_{i}}{\sum }{\tilde{w}}_{ij}^{d^{2}}{\left(y_{ij}-{\hat{P}}_{i}^{HT}\right)}^{2}.$$

This HT estimator falls into the group of considered direct estimators, as they are based only on the area sample data [13,14].

### 2.3. Bayesian Hierarchical Spatial Smoothing Models

Bayesian hierarchical spatial smoothing models have recently gained attention regarding their use in small area estimation instead of direct estimators [4,5]. These methods rely on the assumption that area-specific estimates borrow information from other areas, which makes it possible to find more accurate estimates. Furthermore, this creates the advantage that estimates can be obtained in areas with no samples. The models involve three stages: (i) the likelihood of the response, which is defined conditionally on latent variables (random effects); (ii) the latent variables themselves are given a distribution, and (iii) the specification of prior distributions of all unknown parameters.

A three-stage Bayesian hierarchical spatial rmoothing model for the total number of individuals with the disease in area i given by $y_{i}=\underset{j\in s_{i}}{\sum }y_{ij}$ uses a binomial distribution for stage one as: (7)$$y_{i}|P_{i}∼Binomial\left(n_{i},P_{i}\right),$$

In the second stage, we model the between-area variation in $P_{i}$ using the area random-effects model. In recent times, this has involved incorporating both non-spatial and spatial random effects using the convolution model of Besag–York–Mollié (BYM) [15]. Besides, the standard binomial spatial model, we will describe a series of models based on the BYM model that have been used to perform spatial analyses on the prevalence data from health surveys.

Using the Binomial distribution in (7), our first model is a standard spatial modelling approach for count data using health survey data to estimate the spatial distribution of disease risk. It simply links the estimated prevalence of the disease with the two types of area random terms via a logit function as:(8)$$logit\left(P_{i}\right)=\beta_{0}+u_{i}+v_{i},$$
$$u_{i}|u_{i^{\prime },i\neq i^{\prime }}∼N\left(\frac{1}{a_{i}}\underset{i^{\prime }\in ne\left(i\right)}{\sum }u_{i^{\prime }},\frac{\sigma_{u}^{2}}{a_{i}}\right),$$
$$v_{i}|\sigma_{i}^{2}{∼}_{iid}N\left(0,\sigma_{v}^{2}\right),\;i=1,\ldots ,I,$$
where $\beta_{0}$ is the intercept; $v_{i}$ is the unstructured random component; $u_{i}$ is the structured spatial random component; ne(i) indicates the set of neighbours, and $a_{i}$ is the number of neighbours for a given area i. Here, we adopt the common convention of neighbouring, which considers two areas as neighbours if they share a common boundary. The specification of the structured random effects is based on an intrinsic conditional autoregressive (ICAR) prior [15,16]. We call this, the Binomial Spatial Model, as Model 1.

In the third stage, we require priors for $\beta_{0}$ and the variances of the random effects. The model in Equation (8) results in the smoothing of extreme area estimates in areas with small sample sizes. However, without the incorporation of sampling weights, the estimated of the spatial patterns of disease risk could be biased. 

### 2.4. Incorprating Survey Sampling Weights in Hierarchical Spatial Model Analysis

Following studies [4,5], various approaches have been proposed to account for the survey sampling weight. Let us consider now Models 2 and 3 which are based on the HT estimator. As the HT estimator could be skewed, the estimates are often transformed to approximately conform to normality, Some of the most common transformations are bases the logit and arcsine functions. We first consider the logit transformation. Thus, Model 2 is given by: $$logit\left({\hat{P}}_{i}^{HT}\right)|P_{i}∼N\left(logit\left(P_{i}\right),\sigma_{i}^{2}\right)$$
(9)$$logit\left(P_{i}\right)=\beta_{0}+u_{i}+v_{i},$$
where $logit\left({\hat{P}}_{i}^{HT}\right)$ has variance $\sigma_{i}^{2}=\hat{var}\left({\hat{P}}_{i}^{HT}\right)/{\left({\hat{P}}_{i}^{HT}\left(1-{\hat{P}}_{i}^{HT}\right)\right)}^{2}$. We will call Model 2, the Logit Normal (LN) spatial model. For the arcsine square-root transformation, given as $\arcsin\;\left(\sqrt{{\hat{P}}_{i}^{HT}}\right)$ [5,17], the Arcsine (AS) spatial model leads to the following model specification: $$\arcsin\;\left(\sqrt{{\hat{P}}_{i}^{HT}}\right)|P_{i}∼N\left(\arcsin\;\left(\sqrt{P_{i}}\right),\sigma_{i}^{2}\right)$$
(10)$$\arcsin\;\left(\sqrt{P_{i}}\right)=\beta_{0}+u_{i}+v_{i},$$and the variance of the arcsine transformation is $\sigma_{i}^{2}=\frac{1}{4n_{i}^{E}}$, where $n_{i}^{E}={\hat{P}}_{i}^{HT}\left(1-{\hat{P}}_{i}^{HT}\right)/\hat{var}\left({\hat{P}}_{i}^{HT}\right)$ is the effective sample size in area i. Thus, our Model 3 is the Arcsine square root (AS) spatial model.

For Model 4, we consider pseudo-likelihood (PL), which uses a weighted likelihood [5], where the response values $\left(y_{ij}\right)$ are weighted using the normalised design weights. Thus, rather than using the binomial outcomes used in (7) (Model 1), here, we use $y_{i}^{PL}=\sum_{j\in s_{i}}{\tilde{w}}_{ij}^{d}y_{ij}$ as: $$y_{i}^{PL}|P_{i}∼Binomial\left(n_{i},P_{i}\right)$$
(11)$$logit\left(P_{i}\right)=\beta_{0}+u_{i}+v_{i}.$$

A drawback of the general approach is that the appropriate standard error is not recovered in the case of clustering. Rabe-Hesketh and Skrondal [18] used a pseudo-likelihood method with scaled weights and used sandwich estimation to provide valid standard error estimates within a multilevel framework but did not consider spatial smoothing. We denoted the pseudo-likelihood (PL) spatial model as Model 4. Our Models 5 and 6 are also variations of Model 1, but they now depend on effective sample size and the number of cases. For Model 5, the effective sample size $n_{i}^{E}$ is computed as previously shown and depends on the weighted estimator of prevalence [4]. The effective sample size is the sample size that is required to make the variance under the complex survey design equivalent to that of a simple random sample [4]. Then, the effective number of cases is easily found as $y_{i}^{E}=n_{i}^{E}{\hat{P}}_{i}^{HT}$. As for Model 6, the effective sample size is obtained by using the design effect in area i and is estimated as: (12)$$n_{i}^{E}=\frac{n_{i}}{deff_{i}},$$
where:(13)$$deff_{i}=\frac{s_{i}^{2}}{s_{ri}^{2}}\;for\;i=1,\ldots ,I,$$
where $s_{i}^{2}$ is the unbiased direct estimate of the variance of the sample proportion based on the complex sampling design and $s_{ri}^{2}$ is the unbiased direct estimate of the variance of the proportion based on the simple random sampling design [19]. As before, this resulted in the effective number of cases in area i as $y_{i}^{E}=n_{i}^{E}{\hat{P}}_{i}^{HT}$.

### 2.5. Bayesian Inference, Computation, and Model Evaluation

For the Bayesian estimation of the model parameters, we assumed an improper uniform prior for $\beta_{0}$ and Gamma(0.5,0.008) priors for both the spatial and non-spatial precision parameters $\sigma_{u}^{-2}$ and $\sigma_{v}^{-2}$ as in [5]. The estimations were done using Integrated Nested Laplace Approximation (INLA), which is implemented in the INLA package within the statistical computer software R [20,21,22]. Detailed description of INLA are provided in Appendix A.

Model comparison and selection were carried out using the deviance information criterion (DIC) [23]. DIC value is computed as
$$DIC=\bar{D}+pD,$$
where $\bar{D}$ is the posterior mean of the deviance which measures the goodness of fit and pD is the effective number of parameters which penalises for the complexity of the model. Models with the smallest DIC indicated a better model fit.

## 3. Application

Although Malawi and Mozambique have experienced substantial improvements in child health, preventable child deaths continue to be unacceptably high to achieve the Sustainable Development Goals [24,25,26,27,28,29]. Understanding the local epidemiology of diseases in these two countries is critical for defining and prioritising interventions that can contribute to accelerating the reduction of morbidity and mortality in children under 5 years old in these countries. 

We used Models 1–6, presented above, to estimate the geographical distribution of stunting, wasting, and underweight among children under 5 years old at the district level in Malawi, and childhood fever and diarrhoea at the province level in Mozambique.

### 3.1. Data Sources: Malawi and Mozambique

The 2015-16 MDHS was a national, population-based, cross-sectional survey that was conducted between December 2015 and February 2016. Briefly, the 2015-16 MDHS employed a two-stage sampling designed to produce a nationally representative sample at the national level, residence level (urban and rural), and district level. Stratification was made at two levels: the district level (32 districts), and the urban and rural areas. In the first stage, based on the Malawi Population and Housing Census conducted in Malawi in 2008, and updated based on the General Agriculture Census 2009, 850 primary sample units (PSUs) were selected, which were the enumeration areas (EAs), with a probability proportional to their size (size given by the number of households in each enumeration area). Of these PSUs, 173 were in urban areas and 677 were in rural areas. The second stage of sampling involved a systematic selection of 30 households from each urban cluster and 33 households from each rural cluster, yielding a sample size of 27,516 households from the clusters. The response rate was 99%. The methodology used in the 2015-16 MDHS has been reported in detail in [8]. Figure 1a depicts the geospatial arrangement of the districts of Malawi.

The second dataset used was the IMASIDA 2015, which includes information from 7169 households, interviewing 7749 women aged 15 to 59 years and 5283 men aged 15 to 59 years, over 307 EAs, with data collected between June and September 2015 through a two-stage sampling process designed to produce representative estimates at the national, provincial (11 geographic areas: Maputo Province, Maputo City, Inhambane, Gaza, Sofala, Manica, Zambezia, Nampula, Tete, Niassa, and Cabo Delgado), regional (north, centre and south), and the residence of areas (urban and rural), and for women and men aged 15–59 years. The methodology used has been reported in detail elsewhere [9]. Figure 1b depicts the geospatial arrangement of the provinces of Mozambique. 

All of these datasets are publicly available and can be downloaded at https://dhsprogram.com/ (accessed on 21 July 2021). 

### 3.2. Outcomes

The outcomes considered in this study for childhood in Malawi are three nutritional statuses of children, namely stunting, wasting, and underweight. Anthropometric measurements were used to define the nutritional status of children. Children with a z-score of two standard deviations (−2 SD) below the median of the WHO reference population on height-for-age are categorised as stunted; on weight-for-height as wasted, and on weight-for-age as underweight [24]. Thus, all outcome variables were binary, taking a value of “1” if a child is malnourished (i.e., stunted, wasted, or underweight), and a value of “0” otherwise. Due to missing data on these measurements, only 5149 children were considered for stunting analyses, 5178 for wasting, and 5223 for underweight, respectively. 

For the Mozambique data, the outcomes considered were the fever and diarrhoea statuses. Children under 5 years old who had their mother answer whether they had diarrhoea or fever within the past 2 weeks were included in the analysis. The remaining children with missing values for the outcomes were excluded from our research. Thus, our analyses included a total of 4972 children under 5 years old for fever and 4980 children for diarrhoea. 

### 3.3. Malawi: District Variation in the Prevalence of Child Malnutrition 

The observed prevalence (weighted) of stunting, wasting, and underweight in Malawi among children under 5 years old was 36.82% (95% CI 35.18–38.46), 2.79% (95% CI 2.24–3.33), and 11.58% (95% CI 10.49–12.67), respectively, with the variation across districts that ranged from 15.44% in Mzuzu City to 45.88% in Mchinji for stunting; 1% in Balaka to 9.92% in Nsanje for wasting, and 1.93% in Zomba City to 18.85% in Nsanje for underweight (Table A1 in Appendix B). 

We applied Models 1–6 to estimate the district-level pattern of child growth measures in Malawi. The fit and parameter estimates are presented in Table 1, Table 2 and Table 3. For each child’s growth measurement, the results showed similar estimates for the intercept parameters, except for Model 3 (AS), which was on a different scale. The credible intervals for the intercept parameters were generally narrower when the sampling weights were accounted for. The spatial Model 3 (AS) performed better (DIC = −86.69 for stunting; DIC = −94.76 for wasting; DIC = −89.7 for underweight). 

Figure 2 presents maps of the district-level observed and spatially estimated prevalence of stunting. The spatial pattern in the stunting prevalence was smoother compared to the spatial pattern based on the observed prevalence. Generally higher stunting rates were found in the main central districts of the country. For wasting (Figure 3), districts in the southern part of the country bore the most burden. The spatial trend for wasting was similar to that of underweight prevalence (Figure 4).

### 3.4. Mozambique: Pronvicial Variations in the Prevalence Child Fver and Diarrhoea

A summary of the province’s prevalence of fever and diarrhoea is provided in Table A2 in Appendix B. Overall, about 29.37% (95% CI 26.99–31.87) of children under 5 years old had a fever and 11.11% (95% CI 9.93–12.41) had diarrhoea, with variation across provinces in Mozambique, ranging from 14.37% in Tete to 51.67% in Zambezia for fever, and 6.8% in Tete to 17.19% in Niassa. The model-fit criteria values and parameter estimates are presented in Table 4 and Table 5. The estimates of the intercepts from the models for each condition were similar, except for Model 3 (AS) which was an indifferent scale. Moreover, the estimates of the intercepts were slightly more precise by having narrower credible intervals when sampling weights were accounted for. Furthermore, the spatial Model 3 (AS) was the best fitting model.

Figure 5 and Figure 6 present prevalence maps of fever and diarrhoea, respectively, under different spatial model specifications. Child fever was more concentrated in provinces around the northeastern parts of Mozambique and less in the southern provinces (e.g., Maputo Cidade and Maputo Province). On the other hand, child diarrhoea was higher in most northern provinces, Zambezia, and Niassa provinces, and much lower in the southern parts.

## 4. Discussion and Conclusions

In this paper, we compared several statistical methods and their resulting estimates of spatial distributions of child malnutrition, fever and diarrhoea in Malawi and Mozambique using health survey data, accounting for health survey sampling weights. The results of the study showed that the sampling weight-adjusted methods were the best fitting, a finding similar to previous studies [4,5,6,7]. Even though the estimated spatial pattern was similar, the models that adjusted for the sampling weight produced estimates that had lower variability (narrowed confidence intervals). Thus, using accounting for sampling weights produced estimates of disease risk that had an increased level of confidence. 

There are some concerning issues arising from our study. Firstly, although the arcsine transformation of the weighted prevalence was preferred for our application, it does not have an intuitive interpretation of the association between the binary outcomes and predictors. Secondly, for the Mozambique case, the study used a province which has a much coarser level of geographical aggregation. This may have concealed variations at some higher spatial resolution needed for local policy decisions. We suggest, for future studies, performing spatial analyses at higher spatial resolutions, for example, at the district level. Thirdly, our analyses used univariate spatial methods for the conditions that could be correlated at ecological levels [30,31]. We are now extending these statistical methods to the estimation of joint spatial patterns of diseases. Finally, we did not perform any simulation study to compare the performance of the studied statistical methods for the estimation of spatial disease patterns using complex health survey data. However, we thought that this was not necessary for this study as we aimed to describe the sampling weights adjusting methods and illustrate their use on typical examples. Other previous research work considered their performances using simulations [4,5,6,7]. 

In conclusion, we recommend spatial epidemiology researchers consider incorporating survey sampling weights in disease-mapping analyses for estimating the spatial distribution of disease risks based on complex health survey data. The estimates are more precise, thus providing reliable supporting evidence to drive public health policy on targeting resources in areas of most need. 

## Figures and Tables

**Figure 1 ijerph-19-06319-f001:**
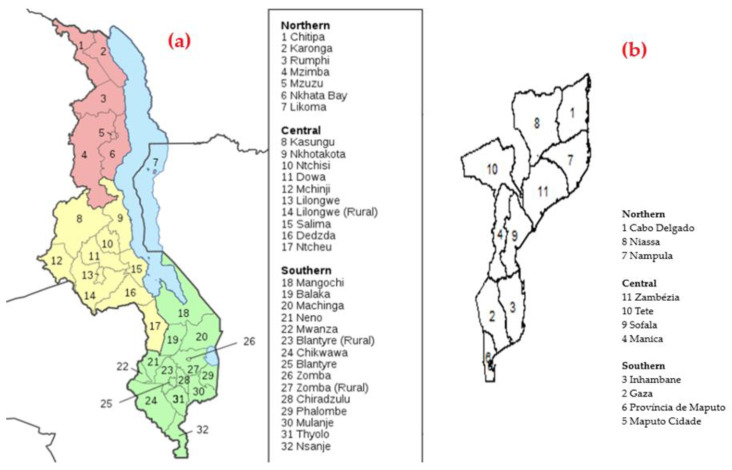
Map of Malawi showing the 32 districts (**a**) and map of Mozambique showing the 11 provinces (**b**).

**Figure 2 ijerph-19-06319-f002:**
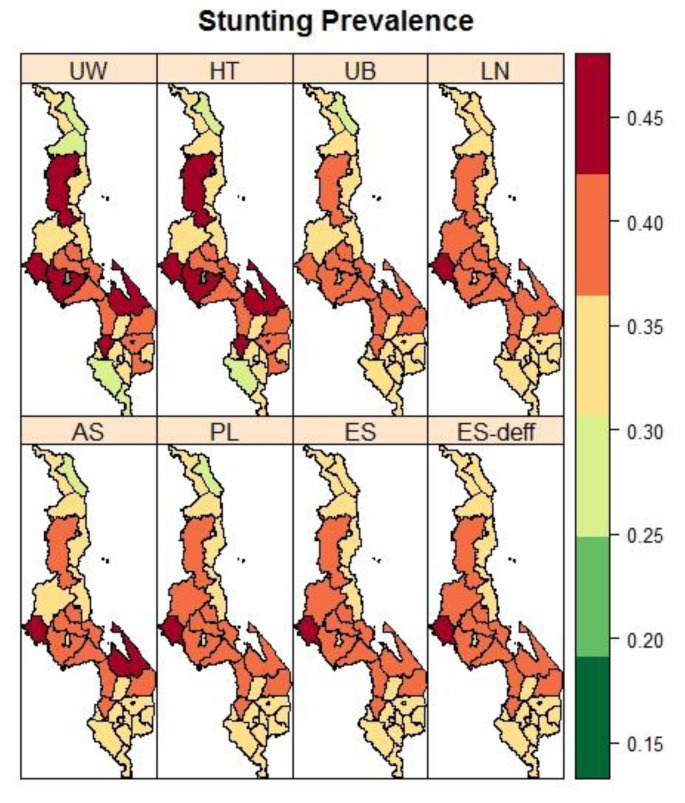
Maps of the observed (UW: Unweighted, HT: Horvitz–Thompson) and spatial estimated prevalences of stunting (UB: Unadjusted Binomial estimator (Model 1), LN: Logit-normal estimator (Model 2), AN: Arcsine-square root transformation estimator (Model 3), PL: Pseudo-likelihood estimator (Model 4), ES: Effective Sample size estimator (Model 5), and ES-deff: Effective Sample size estimator using design effect (Model 6)) by the district in Malawi using 2015-16 MDHS.

**Figure 3 ijerph-19-06319-f003:**
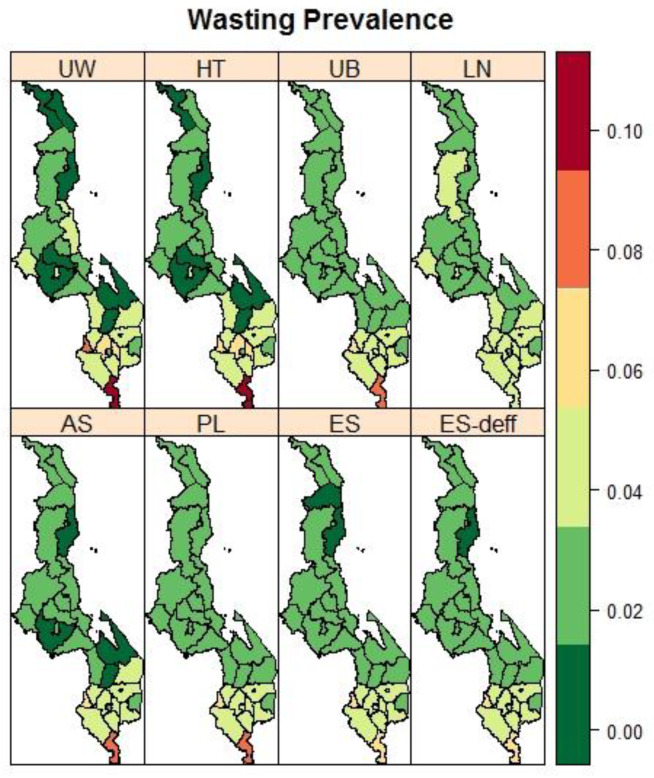
Maps of the observed (UW: unweighted and HT: Horvitz–Thompson) and spatial estimated prevalences of wasting (UB: Unadjusted Binomial estimator (Model 1), LN: Logit-normal estimator (Model 2), AN: Arcsine-square root transformation estimator (Model 3), PL: Pseudo-likelihood estimator (Model 4), ES: Effective Sample size estimator (Model 5), and ES-deff: Effective Sample size estimator using design effect (Model 6)) by the district in Malawi using 2015-16 MDHS.

**Figure 4 ijerph-19-06319-f004:**
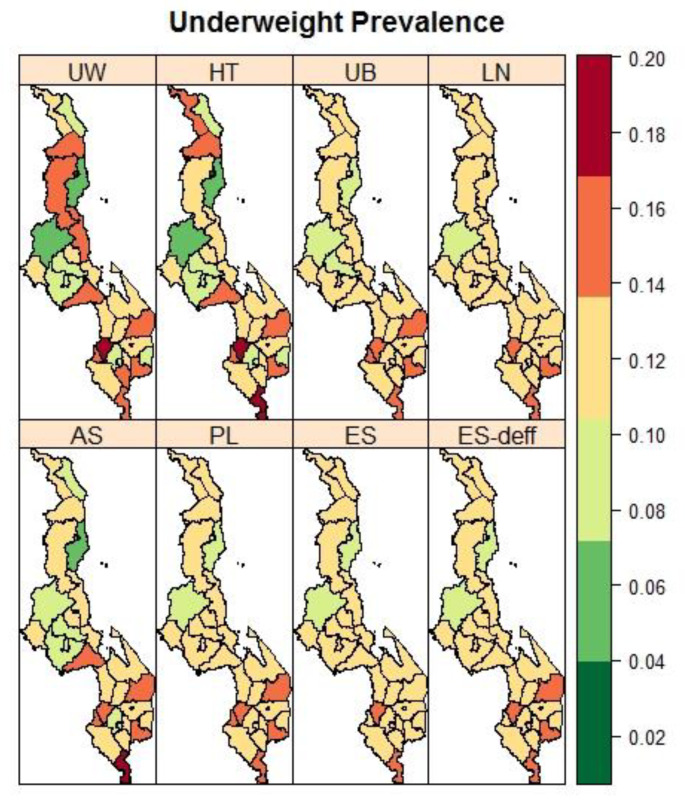
Maps of the observed (UW: unweighted and HT: Horvitz Thompson) and spatial estimated prevalences of underweight (UB: Unadjusted Binomial estimator (Model 1), LN: Logit-normal estimator (Model 2), AN: Arcsine-square root transformation estimator (Model 3), PL: Pseudo-likelihood estimator (Model 4), ES: Effective Sample size estimator (Model 5), and ES-deff: Effective Sample size estimator using design effect (Model 6)) by the district in Malawi using 2015-16 MDHS.

**Figure 5 ijerph-19-06319-f005:**
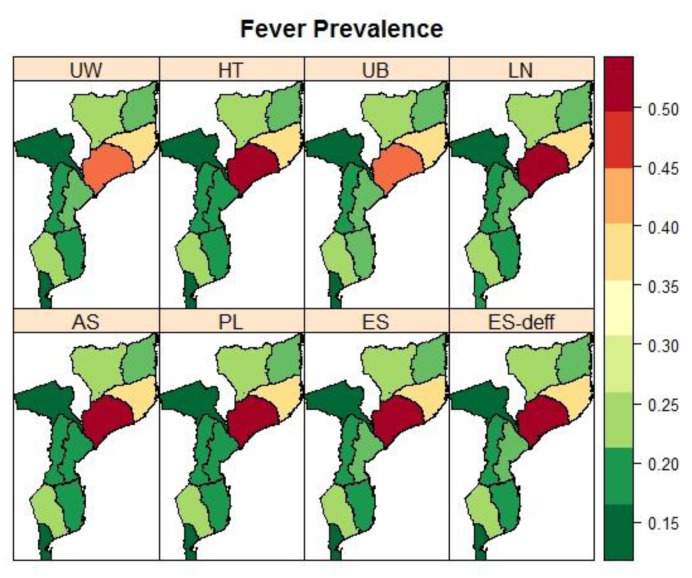
Maps of the observed (UW: unweighted and HT: Horvitz–Thompson) and spatial estimated prevalences of fever (UB: Unadjusted Binomial estimator (Model 1), LN: Logit-normal estimator (Model 2), AN: Arcsine-square root transformation estimator (Model 3), PL: Pseudo-likelihood estimator (Model 4), ES: Effective Sample size estimator (Model 5), and ES-deff: Effective Sample size estimator using design effect (Model 6)) by the province in Mozambique using IMASIDA 2015.

**Figure 6 ijerph-19-06319-f006:**
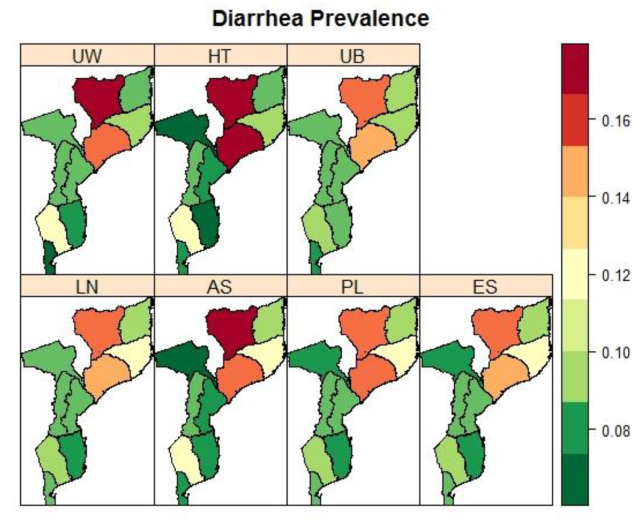
Maps of the observed (UW: unweighted and HT: Horvitz–Thompson) and spatial estimated prevalences of diarrhoea (UB: Unadjusted Binomial estimator (Model 1), LN: Logit-normal estimator (Model 2), AN: Arcsine-square root transformation estimator (Model 3), PL: Pseudo-likelihood estimator (Model 4), and ES: Effective Sample size estimator (Model 5)) by the province in Mozambique using IMASIDA 2015.

**Table 1 ijerph-19-06319-t001:** A comparison of spatial models for mapping child stunting in Malawi using 2015-16 MDHS.

Parameters	Model 1	Model 2	Model 3	Model 4	Model 5	Model 6
$\beta_{0}$ (CI)	−0.605	−0.585	0.634	−0.607	−0.594	−0.609
	(−0.706; −0.547)	(−0.67; −0.505)	(0.608; 0.659)	(−0.691; −0.526)	(−0.68; −0.512)	(−0.694; −0.528)
Sd	0.04	0.042	0.013	0.042	0.043	0.042
$\sigma_{u}$	0.210	0.170	0.069	0.219	0.188	0.218
$\sigma_{v}$	0.114	0.115	0.053	0.125	0.125	0.126
−2LL	−135.88	−24.45	12.88	−137.78	−133.44	−137.21
p(D)	18.54	16.47	23.56	19.55	17.64	19.32
DIC	226.85	6.24	−86.69	229.23	222.80	228.22

**Table 2 ijerph-19-06319-t002:** A comparison of spatial models for mapping child wasting in Malawi using 2015-16 MDHS.

Parameters	Model 1	Model 2	Model 3	Model 4	Model 5	Model 6
$\beta_{0}$ (CI)	−3.514	−3.458	0.166	−3.577	−3.658	−3.609
	(−3.715; −3.325)	(−3.661; −3.257)	(0.143; 0.19)	(−3.78; −3.386)	(−3.87; −3.457)	(−3.814; −3.416)
Sd	0.099	0.103	0.012	0.1	0.105	0.101
$\sigma_{u}^{2}$	0.493	0.353	0.061	0.486	0.519	0.467
$\sigma_{v}^{2}$	0.148	0.121	0.0481	0.144	0.152	0.143
−2LL	−95.22	−48.39	18.11	−94.04	−93.43	−92.31
p(D)	13.94	9.68	22.59	13.13	13.22	12.44
DIC	150.84	60.80	−94.76	148.73	147.06	145.98

**Table 3 ijerph-19-06319-t003:** A comparison of spatial models for mapping child underweight in Malawi using 2015-16 MDHS.

Parameters	Model 1	Model 2	Model 3	Model 4	Model 5	Model 6
$\beta_{0}$ (CI)	−2	−1.981	0.344	−2.003	−2.022	−2.008
	(−2.109; –1.897)	(−2.09; −1.877)	(0.32; 0.368)	(−2.112; −1.901)	(−2.133;−1.916)	(−2.116; −1.905)
Sd	0.054	0.054	0.012	0.053	0.055	
$\sigma_{u}$	0.1390	0.1197	0.0634	0.1523	0.1400	0.1420
$\sigma_{v}$	0.1371	0.1139	0.0495	0.1286	0.1261	0.1321
−2LL	−117.62	−29.98	15.99	−117.38	−113.23	−115.35
p(D)	13.38	10.42	22.91	13.41	12.23	12.91
DIC	197.75	24.96	−89.77	197.28	190.55	193.68

**Table 4 ijerph-19-06319-t004:** A comparison of spatial models for mapping child fever in Mozambique using IMASIDA 2015.

Parameters	Model 1	Model 2	Model 3	Model 4	Model 5	Model 6
$\beta_{0}$ (CI)	−1.137	−1.123	0.523	−1.129	−1.13	−1.13
	(−1.433; −0.844)	(−1.453; −1.123)	(−0.45; 0.597)	(−1.496; −0.767)	(−1.458; −0.805)	(−1.458; −0.805)
Sd	0.146	0.162	0.036	0.183	0.161	
$\sigma_{u}$	0.1734	0.1863	0.1046	0.1854	0.1848	0.1848
$\sigma_{v}$	0.4660	0.5123	0.1094	0.5241	0.5172	0.5172
−2LL	−62.86	−16.20	−0.834	−63.98	−61.28	−61.28
p(D)	10.52	10.28	10.66	10.60	10.50	10.50
DIC	89.60	0.135	−37.64	89.54	86.86	86.86

**Table 5 ijerph-19-06319-t005:** A comparison of spatial models for mapping child diarrhoea in Mozambique using IMASIDA 2015.

Parameters	Model 1	Model 2	Model 3	Model 4	Model 5
$\beta_{0}$ (CI)	−2.145	−2.14	0.329	−1.945	−2.152
	(−2.319;−1.979)	(−2.335; −1.956)	(0.283; 0.374)	(−2.189; −1.708)	(−2.34; −1.975)
Sd	0.085	0.095	0.023	0.118	0.091
$\sigma_{u}^{2}$	0.1651	0.1799	0.0748	0.2303	0.1724
$\sigma_{v}^{2}$	0.2257	0.2231	0.0640	0.3438	0.2442
−2LL	−50.27	−10.40	4.37	−53.70	−48.96
p(D)	8.75	7.90	10.15	9.83	8.64
DIC	81.04	4.30	−39.23	81.71	78.83

## Data Availability

The datasets used in this study are publicly available and can be downloaded at https://dhsprogram.com/ (accessed on 21 July 2021).

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
