# Peer review of "Accounting for Sampling Weights in the Analysis of Spatial Distributions of Disease Using Health Survey Data, with an Application to Mapping Child Health in Malawi and Mozambique"

_ijerph, 2022, doi:10.3390/ijerph19106319_

Round 1

Reviewer 1 Report

The authors provide a timely description of statistical techniques that overcome challenges in statistically analyzing geographically distributed health data. Most importantly, accounting for the sampling weight was shown to reduce variance, therefore increasing confidence, on the conclusions made form the statistical analysis on the data provided. 

The following comments should hopefully help the authors improve the article and better demonstrate the merit of the authors' findings:

1) The abstract of this paper does not clearly state the results, namely that some statistical quantities were similar to previous methods and some were different. It should be made a bit more clear in the abstract what the authors will demonstrate with the article. It may require a few extra sentences in the abstract.

2) Likewise, the introduction does not explain the importance of this research, or even allude to the fact that improved statistical techniques in this research area can help save the lives of children. The context of this research and the intended impacts of the authors should be made more clear in the introduction. 

3) Throughout the article, side-by-side maps are shown to demonstrate the similarity of two methods. By eye, it is time-consuming and a bit painstaking to continuously compare these figures, for example, by comparing small changes in color from province to province between the two figures. If possible, it would be easier for the reader to digest a relative error score or classification score to compare the two methods. Although the maps are interesting and useful, it may be tiring for a reader to digest them all. 

4) If possible, a shortening of the article could be helpful for readers. Reducing the number of figures by following comment 3) could help to do that. 

5) Is there any theoretical backing why variance should be reduced? That would be very helpful for the readers and may be of broader interest if this article is one of the first numerical confirmations of this theoretical result. Some citations in this broader context would be helpful. 

Author Response

Reviewe1

Open Review

Comments and Suggestions for Authors

The authors provide a timely description of statistical techniques that overcome challenges in statistically analyzing geographically distributed health data. Most importantly, accounting for the sampling weight was shown to reduce variance, therefore increasing confidence, in the conclusions made form the statistical analysis on the data provided. 

Response: Thanks for this positive summary of our paper.

The following comments should hopefully help the authors improve the article and better demonstrate the merit of the authors' findings:

1) The abstract of this paper does not clearly state the results, namely that some statistical quantities were similar to previous methods and some were different. It should be made a bit more clear in the abstract what the authors will demonstrate with the article. It may require a few extra sentences in the abstract.

Response: Thanks for this. We thought not to state what the article will demonstrate since we may have not known prior what will result would be from different adjustment methods.

 2) Likewise, the introduction does not explain the importance of this research, or even allude to the fact that improved statistical techniques in this research area can help save the lives of children. The context of this research and the intended impacts of the authors should be made more clear in the introduction. 

Response: We agree the aim of the paper could not have come out well in the Introduction. Our paper is set in the context of health in Africa where health care is plagued by a host of wide-ranging issues, many of which are socio-economic and environmental. Many diseases which are known and researched worldwide are highly prevalent in Africa, where over 90% of all cases and deaths from malaria, today occur in Africa, and over two-thirds of people infected with HIV/AIDS live in Africa. Two recent authoritative reviews (Manda et al, 2020; Haines and Thiart, 2021) have highlighted the impact of spatial statistics on health in the region where the mapping disease prevalence has been used sound spatial statistical principles to present spatial patterns and predictions. Most of these spatial analyses have used health survey data. In this paper, we did not want to bore readers with details of context regarding child illnesses which have extensively been analyzed in several spatial analyses. Rather, we wanted to highlight the shortcoming of many such spatial analyses, and suggest innovative approaches that can be used to perform spatial analysis of health survey data so that mapping disease prevalence in Africa is at least robust to key issues of health. The two datasets used here are purely for illustrative purposes. We have now incorporated some of these in the introduction.

3) Throughout the article, side-by-side maps are shown to demonstrate the similarity of two methods. By eye, it is time-consuming and a bit painstaking to continuously compare these figures, for example, by comparing small changes in color from province to province between the two figures. If possible, it would be easier for the reader to digest a relative error score or classification score to compare the two methods. Although the maps are interesting and useful, it may be tiring for a reader to digest them all. 

Response: We thought presenting maps showing disease risk would be better. We agree that there are so many maps, and we have reduced the number of maps and text in the text.

4) If possible, a shortening of the article could be helpful for readers. Reducing the number of figures by following comment 3) could help to do that. 

Response: We totally agree. We have reduced the number of tables and maps and the text. We also found some maps and Tables were rather redundant or had no meaningful contribution to the overall goals of the paper.

5) Is there any theoretical backing why variance should be reduced? That would be very helpful for the readers and may be of broader interest if this article is one of the first numerical confirmations of this theoretical result. Some citations in this broader context would be helpful. 

Responses:  It is well-known that a weighted mean has a variance larger than an unweighted mean. This is the case for independent observation. However, in Conditional Autoregressive (CAR) models, the full conditional distribution, each area of disease risk is conditional on the sum of the weighted values of its neighbors and has an unknown variance. However, the intrinsic Conditional Auto-Regressive (ICAR) model is a CAR model that assumes a complete spatial correlation between regions and variance  is divided by it neighbors, which entails lower variances than in CAR or non-spatial specifications

 L.M. Haines and C. Thiart, The impact of spatial statistics in Africa. Spatial Statistics (2021), doi:

Reviewer 2 Report

The article studies the spatial variability of health phenomena and their related geospatial data distribution. The authors present six different models for the spatial analysis of these data. They demonstrate the application of their method to two case studies, 'stunting, wasting and underweight among children under-five at the district level in Malawi and childhood fever and diarrhea at the province level in Mozambiqe'.

The basic idea of the paper seems to be worth publishing. However, both the research design and the implementation suffer from major weaknesses. I therefore conclude that this paper can not be published in its current form.

Some justifications for this assessment.

Mixing the two use cases not only does not add value, it moreover adds to the confusion. The authors should focus on one use case, either Malawi or Mozambique. Since the data for Malawi are available at the high resolution district level, this use case seems to be more appropriate. As a higher level of aggregation, the areas highlighted in color in Figure 28 could replace the Mozambique data. This would provide a solid basis for evaluating the properties of the mathematical models used. Otherwise, one is comparing apples to oranges.

The rationale for the selection of the mathematical models is flawed. After the very detailed presentation of the mathematical formulas, a section should be included at the end of Chapter 2 that explains the rationale for the selection of the six models and summarizes the properties of the models.

The description of the data sets is not very comprehensible. Example: 850 PSUs were selected, of which 92 are urban areas, 32 are rural areas. What are all the others? Very many acronyms are used, the meaning of which must first be tediously decoded, via the glossary or other text passages. This greatly impairs their readability.

Subchapter 3.2 Outcomes: the relation of the figures mentioned to subchapter 3.1 remains unclear. Some of the ambiguity could be removed if the study is limited to the data from Malawi and these data are then described more clearly.

Chapter 4 'Discussion and conclusions' is limited to a few comments on the results obtained and possible further developments. An evaluation of the results obtained with the different models is largely absent.

Structure of Text, Figures and Tables.

The headings of almost all Figures should be improved to clarify the content of the Figure. Example: Figures 2 and 3 should be combined into one Figure, similar to the boxplots in Figure 4. The headings could be 'Non Spatial Stunting Prevalence' (left), 'Spatial Stunting Prevalence' (right). 'observed prevalence' and 'estimated prevalence' should be visually separated, the number of models 1 to 6 should be indicated in the Figures. The same comments apply mutatis mutandis to all other Figures.

It remains largely unclear which criteria led to the placement of Tables, Figures, etc. in the main text or in the Appendices. Why do the maps of the study areas (Figure 18, 19) appear in the Appendix and not in the main text? Why do the bulk of the mathematical formula appear in the main body, but some in Appendices B and C? Table A3 seems to be missing.

The meaning of the Tables starting on page 25 can hardly be understood without looking for the meaning of the acronyms and variable names in widely scattered parts of the text.

Author Response

The article studies the spatial variability of health phenomena and their related geospatial data distribution. The authors present six different models for the spatial analysis of these data. They demonstrate the application of their method to two case studies, 'stunting, wasting, and underweight among children under five at the district level in Malawi and childhood fever and diarrhea at the province level in Mozambique.

The basic idea of the paper seems to be worth publishing. However, both the research design and the implementation suffer from major weaknesses. I, therefore, conclude that this paper can not be published in its current form.

Response. We have re-written the introduction to provide more context and justifications. We set our to present methods that have recently been developed to properly account for sampling weights in spatial analysis for mapping health outcomes using survey data. As an illustration, we choose to map the prevalence of child health in two southern African countries. We believe with these changes, we have strengthened the research design and implementation.

Mixing the two use cases not only does not add value, it moreover adds to the confusion. The authors should focus on one use case, either Malawi or Mozambique. Since the data for Malawi are available at the high-resolution district level, this use case seems to be more appropriate. As a higher level of aggregation, the areas highlighted in color in Figure 28 could replace the Mozambique data. This would provide a solid basis for evaluating the properties of the mathematical models used. Otherwise, one is comparing apples to oranges.

Response: We agree that the Mozambique data is mapped at the courser aggregation level, we have alluded to this in the discussion. In this paper, we chose to illustrate mapping health at different resolutions level to show the usability of the spatial methods. This could be replicated at different health levels appropriate for decision-making. As stated above, we have clarified this now in the revised Introduction section.

The rationale for the selection of the mathematical models is flawed. After the very detailed presentation of the mathematical formulas, a section should be included at the end of Chapter 2 that explains the rationale for the selection of the six models and summarizes the properties of the models.

Response: Thank you for this, we have revised Section 2 to have more clarity. Indeed, there are other models, but the spatial models could be seen as the ones that have commonly been used

The description of the data sets is not very comprehensible. Example: 850 PSUs were selected, of which 92 are urban areas, and 32 are rural areas. What are all the others? Very many acronyms are used, the meaning of which must first be tediously decoded, via the glossary or other text passages. This greatly impairs their readability.

Response: Thanks for this. We correct these numbers in section 3.1. In the same section, we remove the acronym MPHC.

Subchapter 3.2 Outcomes: the relation of the figures mentioned in subchapter 3.1 remains unclear. Some of the ambiguity could be removed if the study is limited to the data from Malawi and these data are then described more clearly.

Response: We have now clarified the description in Section 3.1. We have kept Figure 1 which shows the geospatial arrangement of the districts of Malawi and the provinces of Mozambique. However, we have removed several maps that were rather confusing. We have only kept one Figure for each child’s health outcome with maps of observed and estimated prevalence from the observed and estimated prevalence (each of the six methods).

Chapter 4 'Discussion and conclusions' is limited to a few comments on the results obtained and possible further developments. An evaluation of the results obtained with the different models is largely absent.

Response: Thank you for this. We have now revised Section 4.

Structure of Text, Figures, and Tables.

The headings of almost all Figures should be improved to clarify the content of the Figure. Example: Figures 2 and 3 should be combined into one Figure, similar to the boxplots in Figure 4. The headings could be 'Non Spatial Stunting Prevalence' (left), 'Spatial Stunting Prevalence' (right). 'observed prevalence' and 'estimated prevalence' should be visually separated, the number of models 1 to 6 should be indicated in the Figures. The same comments apply mutatis mutandis to all other Figures.

Response: We have reduced the number of figures in the text. Also, we have corrected the headings for clarity

It remains largely unclear which criteria led to the placement of Tables, Figures, etc. in the main text or in the Appendices. Why do the maps of the study areas (Figure 18, 19) appear in the Appendix and not in the main text? Why do the bulk of the mathematical formula appear in the main body, but some in Appendices B and C? Table A3 seems to be missing.

Response: We agree the placement of Figures and Tables was confusing. Now we have back into the main text, the main Tables comparing the fit of the models. Also, maps from only spatial models are retained. The rest are either in the appendices or in the supplementary material.

The meaning of the Tables starting on page 25 can hardly be understood without looking for the meaning of the acronyms and variable names in widely scattered parts of the text.

Response: As stated before, we have now organized our Tables and Figures for more clarity and flow.

Reviewer 3 Report

The authors applied and compared baseline methods to recently published methods for complex survey sampling to two use cases in two countries using standard performance criteria. The use cases, growth measures and fever and diarrhea, are useful and important for public health policy. The findings were consistent with prior studies, suggesting that these results would contribute towards the accumulated evidence towards generalize-ability and suitability of these methods in new use cases.

Minor Issues:

  1. The coloring of your figures (red, green) is problematic from a red-green color-blind perspective, as well as for printing in grayscale. Optimizing the color scheme will assist future readers.
  2. Be consistent. There seems to be a double-labeling in the figures/text/table (e.g., UB: Unadjusted Binomial estimator (Model 1)) where UB and Model 1 are used interchangeably and inconsistently. I would just get rid of the “Model #” throughout to make this more consistent, as it is not needed and makes comparison more challenging (e.g., Figures use UB whereas table uses Model 1 and text uses both). All abbreviations need to be spelled out in all figure/table caption, when used.
  3. Context is missing in terms of an overall population N (perhaps # who met inclusion criteria?) from Appendix D.1. Table A1 and Appendix D.2. Table A2 as the total has a % that is not 100%. Suggest including the 5149 stunting, 5178 wasting, 5223 underweight, 4972 fever and 4980 diarrheas to provide that missing context.

Author Response

The authors applied and compared baseline methods to recently published methods for complex survey sampling to two use cases in two countries using standard performance criteria. The use cases, growth measures and fever and diarrhea, are useful and important for public health policy. The findings were consistent with prior studies, suggesting that these results would contribute towards the accumulated evidence towards generalize-ability and suitability of these methods in new use cases.

Response: Thanks for this positive summary of our paper.

Minor Issues:

  1. The coloring of your figures (red, green) is problematic from a red-green color-blind perspective, as well as for printing in grayscale. Optimizing the color scheme will assist future readers.

Response: We thought our coloring is somehow standard, so we have opted to keep it that way.

  1. Be consistent. There seems to be a double-labeling in the figures/text/table (e.g., UB: Unadjusted Binomial estimator (Model 1)) where UB and Model 1 are used interchangeably and inconsistently. I would just get rid of the “Model #” throughout to make this more consistent, as it is not needed and makes comparison more challenging (e.g., Figures use UB whereas table uses Model 1 and text uses both). All abbreviations need to be spelled out in all figure/table caption, when used.

Response: Thanks for this. We have now described all abbreviations in the captions.

  1. Context is missing in terms of an overall population N (perhaps # who met inclusion criteria?) from Appendix D.1. Table A1 and Appendix D.2. Table A2 as the total has a % that is not 100%. Suggest including the 5149 stunting, 5178 wasting, 5223 underweight, 4972 fever and 4980 diarrheas to provide that missing context.

Response: We have now included the overall population in Appendix D1 Table A1 and in the appendix D2 Table A2

Round 2

Reviewer 2 Report

The authors have made some efforts to improve their paper. On a positive note, reducing the number of Figures to the essentials improves the readability of the paper. Some additions in the conclusion section are helpful to put the results in a broader context. However, there is also a negative side, as the revision was not always done carefully. There are quite a number of misspellings, (very) poor phrasing, incorrect numbering, etc.

Some examples

line 110: ‚... variation in <blank> using two options ...‘ <blank> is to be replaced by the correct content

lines 135, 180: the numbering of subheadings should be 2.4, 2.5 (instead of 2.2, 2.3), right?

It is now even more difficult to follow the description of Models 1 to 6 than it was in the original manuscript. It seems to me indispensable to include another short subchapter 2.6 comparing the models in a succinct concise form.

lines 198, 199, 200:

One concrete example of the need to improve the language: ‘According to [50], a difference of 3 or less units in DIC values between two models cannot be distinguished, while for a difference of between 3 and 7 units can be weakly differentiated.’ This statement could indicate that DIC values of different models must differ by at least a value of 3 to be considered statistically significant. ‘Weakly differentiated’ could mean that statistical significance is low for values between 3 and 7. This statement needs to be reworded to be comprehensible.

line 244: caption is present, but Figure is missing

Captions Figures 2,3,4, 7, 8: ‘Model 4’ appears twice, is this correct? Figures 5, 6 are not present

Lines 382ff: this paragraph needs special linguistic revision. Currently it is barely understandable. What does ‘course level’ mean? ‘Coarse level’? What are ‘masked variations’? Etc.

Author Response

The authors have made some efforts to improve their paper. On a positive note, reducing the number of Figures to the essentials improves the readability of the paper. Some additions in the conclusion section are helpful to put the results in a broader context. However, there is also a negative side, as the revision was not always done carefully. There are quite a number of misspellings, (very) poor phrasing, incorrect numbering, etc.

Response: We are thankful that our revisions to the previous version have improved the paper. We agree that the last were many and major, and some unintended omissions and errors could have been introduced. However, these were not so major as to distort the overall message and outlook of the paper. In the present version, we have undertaken to thoroughly edit the paper for clarity in technicalities and language

Some examples

line 110: ‚... variation in <blank> using two options ...‘ <blank> is to be replaced by the correct content

Thank you for this, revised.

lines 135, 180: the numbering of subheadings should be 2.4, 2.5 (instead of 2.2, 2.3), right?

Thank you for this, corrected.

It is now even more difficult to follow the description of Models 1 to 6 than it was in the original manuscript. It seems to me indispensable to include another short subchapter 2.6 comparing the models in a succinct concise form.

We have had a closer re-look at the description of the models. We have done some slight corrections, and we think the description is now clearer. We also feel that our present description of the performance of the models per the Table showing the model fits is adequate.

lines 198, 199, 200:

One concrete example of the need to improve the language: ‘According to [50], a difference of 3 or less units in DIC values between two models cannot be distinguished, while for a difference of between 3 and 7 units can be weakly differentiated.’ This statement could indicate that DIC values of different models must differ by at least a value of 3 to be considered statistically significant. ‘Weakly differentiated’ could mean that statistical significance is low for values between 3 and 7. This statement needs to be reworded to be comprehensible.

We agree that this sentence was very vague. We have re-looked at it, and we felt it is not necessary and does not add any value to the overall aims of the paper. Thus, these parts have been removed from the paper.

line 244: caption is present, but Figure is missing

Thank you for this. The figure has now been inserted.

Captions Figures 2,3,4, 7, 8: ‘Model 4’ appears twice, is this correct? Figures 5, 6 are not present

Thank you for this, the caption is now corrected. Figures 7 and 8 are now Figures 5 and 6.

Lines 382ff: this paragraph needs special linguistic revision. Currently it is barely understandable. What does ‘course level’ mean? ‘Coarse level’? What are ‘masked variations’? Etc.

Thank you for this. It was a typo, we meant coarser spatial resolution, which is just lower spatial resolution. We have now used the former terminology. Mask here is used to mean cover or conceal variations at high spatial resolution. We have edited the lines to reflect these better.